# Effect of linker on the binding free energy of stapled p53/HDM2 complex

**Haeri Im, Sihyun Ham** *

Department of Chemistry, The Research Institute of Natural Sciences, Sookmyung Women's University, Seoul, Korea

* sihyun@sookmyung.ac.kr

## Abstract

Inactivation of the tumor suppressor p53 resulting from the binding with a negative regulator HDM2 is among the predominant defects in human cancers. p53-mimicking peptides whose conformational and proteolytic stability is enhanced by an all-hydrocarbon staple are being recognized as promising anticancer agents for disrupting the p53–HDM2 binding and reactivating p53. Herein, we conduct a computational modeling and thermodynamic characterization of stapled p53/HDM2 complex via molecular docking, simulations, and binding free energy analysis. The binding thermodynamics analysis is done based on the end-point calculation of the effective binding energy—a sum of the direct peptide–protein interaction energy and the dehydration penalty—and on its decomposition into contributions from specific groups constituting the complex. This allows us to investigate how individual amino acids in the stapled p53 and HDM2 contribute to the binding affinity. We find that not only the epitope residues (F19, W23 and L26), but also the hydrocarbon linker of the stapled p53 impart significant contributions. Our computational approach will be useful in designing new stapled peptides in which the staple location is also optimized to improve the binding affinity.

## Introduction

The p53 protein is a transcription factor regulating cell cycle and apoptosis in response to DNA damages and cellular stresses [1]. It plays a critical role for maintaining the genome integrity and preventing the development of tumor [2]. The p53 levels in normal cells are controlled by negative regulator proteins such as HDM2 that neutralize the p53 transactivation activity via a direct binding interaction [3]. The loss of p53 activity resulting from the binding with overexpressed HDM2 is among the predominant defects in human cancers [4]. The reactivation of p53 by perturbing the p53–HDM2 binding has therefore been regarded as a promising approach for suppressing tumor growth in cancer cells [5–7].

The p53–HDM2 interaction occurs primarily between the short α-helical segment of p53 and the hydrophobic pocket of the HDM2 surface. In particular, three residues (F19, W23 and L26) within this segment form critical contacts with HDM2 [8]. p53-mimicking peptides incorporating these epitope residues will hence serve as potential anticancer agents that reactivate p53 by driving it out from the interaction with HDM2. In this context, there is recently a

**Data Availability Statement:** All relevant data are within the manuscript and its Supporting Information files.

**Funding:** This study was funded by the Samsung Science and Technology Foundation under Project

Number SSTF-BA1401-52 and the National
Research Foundation of Korea (NRF) (No. NRF-
2017R1A2B3010053) to SH. The funders had no
role in study design, data collection and analysis,
decision to publish, or preparation of the
manuscript.

**Competing interests:** The authors have declared
that no competing interests exist.

growing interest in stapled peptides in which an α-helical conformation appropriate to binding with HDM2 is rigidified through an introduction of an all-hydrocarbon linkage connecting helix forming residues [9–13]. Furthermore, it has been demonstrated that the stapling also enhances the proteolytic stability and promotes cell permeability, which are crucial for *in vivo* therapeutic activity [14]. Systematic mutational analysis, including the optimization of the staple locations, are being carried out experimentally and computationally in search of higher binding affinity and more improved potency and specificity [15,16].

Herein, we propose a computational method that is useful for designing new stapled p53-mimicking peptides (to be simply referred to as stapled p53 peptides from here on). We start from a peptide sequence and conduct a template-based modeling using an experimental structure. The stapled p53 peptide so constructed is subjected to molecular dynamics simulations to explore a representative structure in an aqueous solution and its conformational stability. The simulated stapled p53 structure is then used for a molecular docking onto the HDM2 surface. Starting from the docked complex structure, we perform molecular dynamics simulations. For the simulated complex configurations, we finally carry out thermodynamic analysis. This is done based on the end-point calculation of the effective binding free energy $\Delta f$ [17]. It comprises the direct peptide-protein interaction energy ($\Delta E_u$) and the solvation free energy contribution ($\Delta G_{solv}$), $\Delta f = \Delta E_u + \Delta G_{solv}$. The quantity $\Delta f$ is connected to the binding free energy ($\Delta G_{bind}$) via $\Delta G_{bind} = \Delta f - T(\Delta S_{config} + \Delta S_{ext})$ in which $\Delta S_{config}$ and $\Delta S_{ext}$ are the configurational and external entropies, respectively [18,19]. Since these entropy terms are typically negative, the favorable contributions to the binding affinity arise mainly from $\Delta f$. The formation of peptide–protein contacts, such as hydrogen bonds and van der Waals contacts, leads to favorable changes in the direct interaction energy ($\Delta E_u < 0$). However, the formation of these peptide–protein contacts involves the dehydration penalty ($\Delta G_{solv} > 0$). Therefore, in arguing the net contribution to the binding affinity, it is essential to analyze $\Delta f$ that simultaneously takes into account both $\Delta E_u$ and $\Delta G_{solv}$. Importantly, $\Delta f$ can be decomposed into contributions from specific groups constituting the complex [20,21]. Thereby, our method allows us not only to investigate the relevance of individual amino acids, but also to quantify the contribution from the hydrocarbon linker to the binding affinity.

## Materials and methods

### Modeling stapled p53/HDM2 complex

We investigated a stapled p53 peptide, referred to as sMTide-02 in Ref. [14], whose sequence is Ac-TSFXEYWALLX-NH2 (X: linker positions): two additional peptides stapled at shifted positions, Ac-TXFAEYWAXLS-NH2 (to be referred to as sMTide-02b) and Ac-XSFAEYWXLLS-NH2 (sMTide-02c) were also studied to analyze the dependence of the structural stability and binding affinity on the linker location (see Fig 1A). Each peptide structure was constructed using an experimental structure for a stapled p53 (PDB ID 3V3B) [15] as a template. The hydrocarbon linker was added with GaussView [22]. A 100 ns molecular dynamics simulation was carried out for each stapled peptide starting from the constructed structure (see below for details on the simulation). A representative peptide conformation for each system was selected based on the *k*-means clustering with a radius of 4.0 Å (shown in Fig 1A), which was then used in the docking onto the HDM2 surface. We employed the X-ray structure (PDB ID 1YCR) [8] for HDM2 (Fig 1B). The docking was performed using Auto-Dock Vina [23]. Only the side chains were permitted to rate in the docking simulation. 1,000 complex structures were generated from the docking carried out for each system, and we chose the most stable structure. Molecular dynamics simulations for the stapled p53/HDM2 complexes were then carried out starting from the respective docked structures.

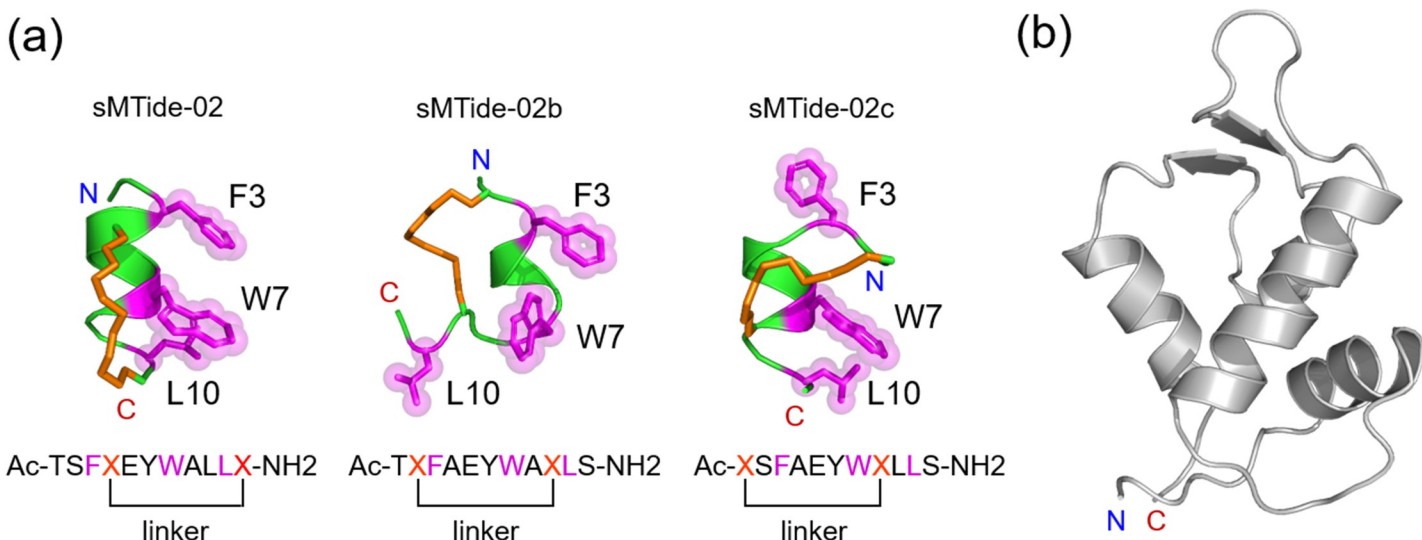

**Fig 1. Stapled p53 peptide and HDM2 structures.** (a) Simulated stapled p53 peptides (sMTide-02, sMTide-02b and sMTide-02c). Three epitope residues (F3, W7, and L10; colored magenta) and the hydrocarbon linker (colored orange) are indicated by stick representations. (b) X-ray structure for HDM2 (PDB ID 1YCR).

## Molecular dynamics simulations

Simulations for the free stapled p53 peptides and the stapled p53/HDM2 complexes were carried out using AMBER16 [24]. Each system was solvated by waters and counter ions. The ff99SB-ILDN [25,26] was adopted for the peptide/protein and for water the TIP3P model [27] was used. The linker partial charges were determined from the restrained electrostatic potential (RESP) method [28] after carrying out an HF/6-31G* quantum mechanical calculation with Gaussian 09 [29]. We employed the general Amber force field [30] for the other parameters. After the standard minimization and equilibration steps, 100 ns *NPT*-ensemble simulations were performed at 300 K and 1 bar. A single and 10 independent runs were conducted for the free peptide and the complex, respectively.

## Thermodynamic calculations

The effective binding free energy, $\Delta f = \Delta E_{\mathrm{u}} + \Delta G_{\mathrm{solv}}$, was computed using the simulated complex structures saved with a 1 ns interval. The direct peptide-protein interaction energy ($\Delta E_{\mathrm{u}}$) can be obtained easily from the force field parameters. The solvation term ($\Delta G_{\mathrm{solv}}$) comprises the ones for the complex and its components, $\Delta G_{\mathrm{solv}} = G_{\mathrm{solv;\ complex}} - (G_{\mathrm{solv;\ stapled\ p53}} + G_{\mathrm{solv;\ HDM2}})$, and was computed using the 3D-RISM theory [31,32].

Within the classical force field, the direct interaction energy ($\Delta E_{\mathrm{u}}$) is expressed as a sum of atomic contributions. For the solvation free energy ($G_{\mathrm{solv}}$), we have recently developed an exact atomic decomposition method [20,21]. Thereby, the effective binding free energy ($\Delta f$) can be partitioned into contributions from constituent atoms. By an appropriate grouping of these atomic terms, individual residue and linker contributions to $\Delta f$ can be obtained.

## Results and discussion

We first performed a 100 ns free-peptide simulation for each of the stapled p53 peptides (sMTide-02, sMTide-02b and sMTide-02c) to examine its conformational stability in an aqueous environment. The free sMTide-02 was quite stable (the Cα root-mean-squared deviation (RMSD) to the initial structure remained <1.0 Å) during the simulation. Its overall helical

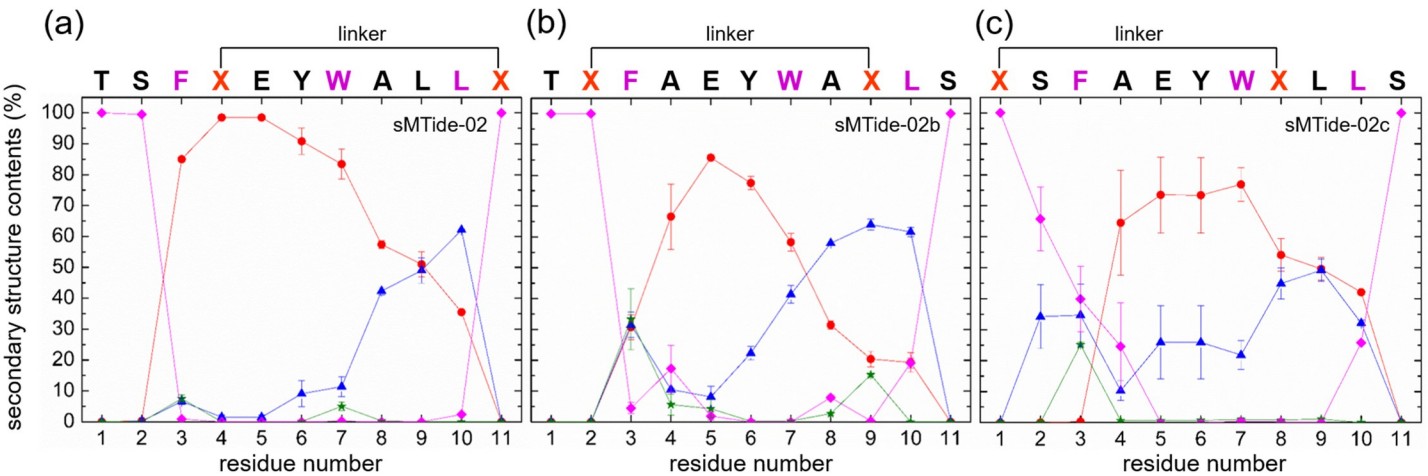

**Fig 2. Average secondary structure contents of the free stapled p53 peptides.** (a–c) Average secondary structure contents for the free sMTide-02 (a), sMTide-02b (b) and sMTide-02c (c) (red circles, helix; blue triangles, turn; green stars, bend, and magenta diamonds, coil).

structure was also maintained (Fig 2A) with an average helical content of 55%. It is well known that short helical segments taken out from globular proteins cannot keep up its secondary structure when isolated [33]. Our simulation thus confirms the significantly enhanced conformational stability of a short helical peptide brought about by the staple linker. On the other hand, the structures of the other stapled peptides (sMTide-02b and sMTide-02c) exhibited somewhat larger deviations: the Cα RMSDs to the respective initial structures increased up to 1.6 Å and 2.4 Å and the average helical contents dropped to 35% (Fig 2B) and 39% (Fig 2C), respectively. Thus, the stability and helicity of the stapled p53 peptides depend on the linker positions.

We next performed 10 independent 100 ns molecular dynamics simulations for the stapled p53/HDM2 complex (formed with each of sMTide-02, sMTide-02b and sMTide-02c) after docking the simulated free stapled p53 peptide onto the HDM2 surface. Representative complex structures therefrom are displayed in Fig 3. In the wild-type p53/HDM2 complex, the epitope residues (F19, W23 and L26) enter deep inside the HDM2 hydrophobic pocket (comprising M26, L30, L33, G34, I37, M38, Y43, H49, V51, F67, V69, H72, I75 and Y76). Such an interface topology is well conserved in the simulated sMTide-02/HDM2 complex; the epitope residues (F3, W7 and L10) of sMTide-02 make hydrophobic contacts with most of the HDM2 residues listed above (L30, F31, L33, G34, I37, M38, Y43, V51, V69, H72, and I75). We also find that the hydrocarbon linker (colored orange) makes contacts with the hydrophobic residues (F31, G34, and M38) in the HDM2 binding surface (colored yellow). Such an interaction between the peptide staple and protein has been observed in previous simulation studies for the related systems [34,35]. For the complexes formed with the stapled p53 peptides in which the linker positions are shifted (sMTide-02b and MTide-02c), we find that the binding of the epitope residues and the linker to HDM2 is not so prominent and involves less contacts with HDM2 (sMTide-02b contacts only with L30, F31, L33, F34, I37, M38, Y43, V69, I75, whereas sMTide-02c with L30, F31, L33, I37, M38, F67, H72, I75). This indicates that the less helical structures of these stapled p53 peptides are not optimal in binding to HDM2.

Finally, we conducted the end-point calculations of the effective binding free energy ($\Delta f$) based on the simulated complex structures. As we stated above, $\Delta f$ provides the major favorable contribution to the binding affinity. Therefore, its decomposition into specific group contributions enables us to identify critical residues. The effective binding free energies for HDM2

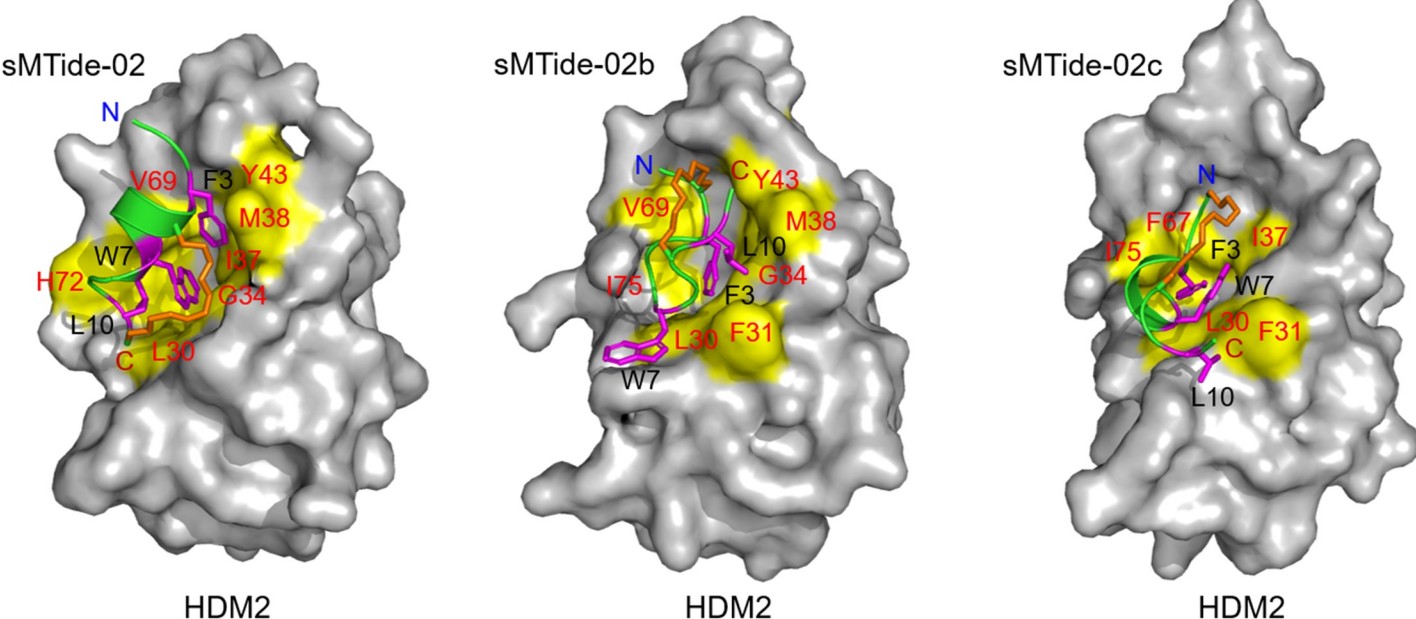

**Fig 3. Representative structures of simulated stapled p53/HDM2 complexes.** Three epitope residues (F3, W7, and L10; colored magenta) of the stapled p53 peptides are indicated by stick representations. The yellow surface shows the hydrophobic surface (comprising the residues colored red) of HDM2 interacting with the epitope residues.

with sMTide-02, sMTide-02b and sMTide-02c were computed to be −18.7 ± 0.5 kcal/mol, −13.2 ± 1.0 kcal/mol, and −7.0 ± 0.8 kcal/mol, respectively (average ± standard error estimated from the respective 10 independent runs). This trend is in accord with the experimental observation for the related SAH-p53 (stabilized alpha-helix of p53) peptides [9]: The linker positions in sMTide-02, sMTide-02b and sMTide-02c corresponds to those in SAH-p53-8, SAH-p53-2 and SHA-p53-3 peptides, respectively, whose binding affinities for HDM2 decrease in this order. The decomposition of the effective binding energy into constituent amino acids is shown in Fig 4. We find that the epitope residues (F3, W7 and L10; colored magenta) of the stapled peptides and the hydrophobic residues located at the HDM2 binding surface (colored yellow) are in fact the principal contributors to the binding affinity. Interestingly, we observe that the hydrocarbon linker of sMTide-02 (colored orange) also provides a significant contribution. In this regard, we emphasize the more relevance of analyzing $\Delta f$ than just examining the direct peptide–protein interaction energy ($\Delta E_u$). Indeed, as can be inferred from S1 Fig that further partitions $\Delta f$ into $\Delta E_u$ and $\Delta G_{solv}$ terms, the contributions from the epitope residues and the hydrocarbon linker to $\Delta E_u$ are comparable to those from the other residues in the stapled peptide, and hence, their significance cannot be elucidated solely in terms of $\Delta E_u$. Only after taking into account the dehydration penalty embodied in $\Delta G_{solv}$, the special role of the epitope residues and the hydrocarbon linker becomes evident. This demonstrates the essential importance of analyzing $\Delta f$ in identifying the residues critical to binding. The significant contributions from the hydrocarbon linker is also observable in sMTide-02b and sMTide-02c, albeit to a lesser extent. (The partitioning of $\Delta f$ into $\Delta E_u$ and $\Delta G_{solv}$ for sMTide-02b and sMTide-02c is presented in S2 and S3 Figs, respectively.) Correspondingly, the contributions from the epitope residues of sMTide-02b and sMTide-02c are somewhat smaller than those of sMTide-02. This indicates that, in designing stapled peptides, the location of the linker should be optimized also from the standpoint of binding affinity.

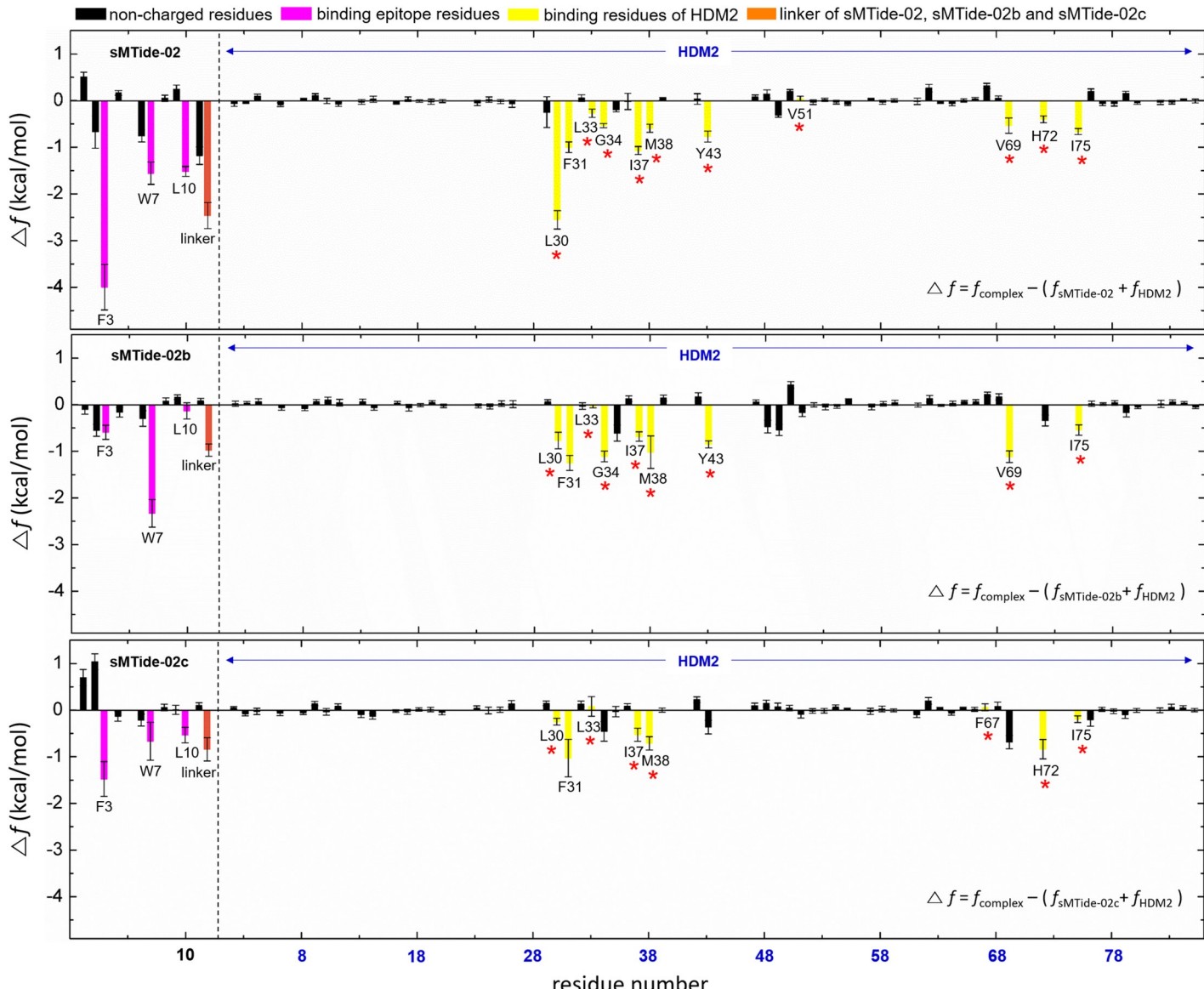

**Fig 4. Analysis of the effective binding free energy (Δf).** The epitope residues of the stapled p53 peptides are colored magenta and the hydrocarbon linker is colored orange; hydrophobic residues located at the binding interface of HDM2 are colored yellow; and residues present in the wild-type p53/HDM2 binding interface are indicated by the red stars.

## Conclusions

We present a computational method for modeling and characterizing stapled peptides and illustrate its application to a stapled p53/HDM2 complex. Putative initial structures for the free stapled p53 peptides and their complexes with HDM2 are generated by a template-based modeling and docking tools, which are subsequently validated via molecular dynamics simulations in an aqueous environment. Thermodynamic characterization of the stapled p53/HDM2 complex is done by decomposing the effective binding free energy into specific constituent groups. This method allows one to identify hot spot residues critical to binding. In fact, we identify the epitope residues of the stapled p53 and the hydrophobic residues of the HDM2 surface to be the principal contributors to the binding affinity. We also find that the

hydrocarbon linker of the stapled p53 provides a significant contribution. Thus, the linker plays an important role not only in stabilizing the helical peptide appropriate to binding, but also in determining the binding thermodynamics. Our method will be useful in designing new stapled peptides in which the staple location can be optimized from the thermodynamic viewpoint.

## Supporting information

**S1 Fig. Further partitioning of the residue-decomposed effective binding energy $\Delta f$ into the $\Delta E_u$ and $\Delta G_{solv}$ terms for the complex with sMTide-02.**
(TIF)

**S2 Fig. Further partitioning of the residue-decomposed effective binding energy $\Delta f$ into the $\Delta E_u$ and $\Delta G_{solv}$ terms for the complex with sMTide-02b.**
(TIF)

**S3 Fig. Further partitioning of the residue-decomposed effective binding energy $\Delta f$ into the $\Delta E_u$ and $\Delta G_{solv}$ terms for the complex with sMTide-02c.**
(TIF)

## Author Contributions

**Conceptualization:** Sihyun Ham.

**Data curation:** Haeri Im, Sihyun Ham.

**Formal analysis:** Haeri Im, Sihyun Ham.

**Funding acquisition:** Sihyun Ham.

**Investigation:** Haeri Im, Sihyun Ham.

**Methodology:** Haeri Im, Sihyun Ham.

**Project administration:** Sihyun Ham.

**Resources:** Sihyun Ham.

**Software:** Sihyun Ham.

**Supervision:** Sihyun Ham.

**Validation:** Sihyun Ham.

**Visualization:** Haeri Im, Sihyun Ham.

**Writing – original draft:** Haeri Im, Sihyun Ham.

**Writing – review & editing:** Haeri Im, Sihyun Ham.

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
