## [Decision Letter · Decision Letter 0]

23 Jan 2020

PONE-D-19-35957

Effect of linker on the binding free energy of stapled p53/HDM2 complex

PLOS ONE

Dear Professor Ham,

Thank you for submitting your manuscript to PLOS ONE. After careful consideration, we feel that it has merit but does not fully meet PLOS ONE’s publication criteria as it currently stands. Therefore, we invite you to submit a revised version of the manuscript that addresses all the points raised by both reviewers.

We would appreciate receiving your revised manuscript by Mar 08 2020 11:59PM. To enhance the reproducibility of your results, we recommend that if applicable you deposit your laboratory protocols in protocols.io, where a protocol can be assigned its own identifier (DOI) such that it can be cited independently in the future. For instructions see: http://journals.plos.org/plosone/s/submission-guidelines#loc-laboratory-protocols

We look forward to receiving your revised manuscript.

Kind regards,

Claudio M. Soares, Ph.D

Academic Editor

PLOS ONE

Journal Requirements:

Reviewers' comments:

Reviewer's Responses to Questions

**Comments to the Author**

1. Is the manuscript technically sound, and do the data support the conclusions?

Reviewer #1: Partly

Reviewer #2: Yes

2. Has the statistical analysis been performed appropriately and rigorously? 

Reviewer #1: No

Reviewer #2: Yes

3. Have the authors made all data underlying the findings in their manuscript fully available?

Reviewer #1: Yes

Reviewer #2: Yes

4. Is the manuscript presented in an intelligible fashion and written in standard English?

Reviewer #1: Yes

Reviewer #2: Yes

5. Review Comments to the Author

Reviewer #1: The manuscript closely replicates the work of Thomas et al (Cell Cycle 2010,9,4560-4568). Although the authors have cited this particular work, they do not discuss the difference, if any, in their approach and results. The conclusions presented also do not add to our present knowledge, as it is already well known that F19, W23, L26 and the hydrocarbon staple are the most important residues in the binding of stapled p53 peptides to HDM2. The authors claim that their "method will be useful in designing new stapled peptides in which the staple location can be optimized from the thermodynamic viewpoint". However, they only did simulations of a single stapled peptide and did not compare it with peptides stapled at other positions. Although ten simulations were performed, standard deviations are not shown in Figures 2 and 4. In light of the lack of new results and conclusions, I am hesitant to accept this manuscript for publication.

Reviewer #2: This manuscript presents the results of molecular dynamics simulation and end-point effective free energy calculations for a stapled peptide interacting with HDM2. The main finding of the paper is that, in addition to the known epitope residues, the hydrocarbon linker imparts a significant contribution to the binding affinity. Overall, the paper meets the standards of publication of PLOS One. Below I have a few comments that, when addressed, would improve the manuscript as well as place it better within the prior literature.

1. As the “effect of linker” is a key area of concentration of this paper, it would be advisable to add some additional calculations and/or comparison of HDM2/peptide interactions where a linker is not present.

2. More detail describing the effective binding free energy calculation should be added to the abstract, as “binding free energy analysis” encompasses many very different techniques.

3. The “Stapled-p53” shorthand that is used throughout can be more clearly defined as the peptide is based upon, but is not, p53. For example, the peptide name used in Ref. 10 for this sequence (sMTide-02) can be utilized.

4. Simulation work on closely related systems should be cited. There are several works that study binding thermodynamics and kinetics of MDM2/HDM2 and p53-like peptides (see e.g. [1] Zwier, et al J. Phys. Chem. Lett. 2016, 7, 3440. [2] Paul, et al. Nature Comm. 2017, 8 1095, [3] Morrone, et al. J. Chem. Theory Comput. 13 863, [4] Morrone, et al. J. Chem. Theory Comput. 2017 13 870, [5] Tran, et al . J. Phys. Chem. B 2019 123 2469.) Specifically, #3 simulates stapled peptides and finds that the peptide staple may interact with the protein.

6. PLOS authors have the option to publish the peer review history of their article (what does this mean?). If published, this will include your full peer review and any attached files.

Reviewer #1: No

Reviewer #2: No

---

## [Author Response · Author response to Decision Letter 0]

5 Mar 2020

Response to Reviewers is described in the "response_letter.docx" uploaded.

---

## [Decision Letter · Decision Letter 1]

19 Mar 2020

PONE-D-19-35957R1

Effect of linker on the binding free energy of stapled p53/HDM2 complex

PLOS ONE

Dear Professor Ham,

Thank you for submitting your manuscript to PLOS ONE. Reviewer #1 present strong arguments for rejecting this manuscript. I will allow a new round of revision to allow you to address the reviewer's criticism in a detailed manner.

We would appreciate receiving your revised manuscript by May 03 2020 11:59PM. To enhance the reproducibility of your results, we recommend that if applicable you deposit your laboratory protocols in protocols.io, where a protocol can be assigned its own identifier (DOI) such that it can be cited independently in the future. For instructions see: http://journals.plos.org/plosone/s/submission-guidelines#loc-laboratory-protocols

We look forward to receiving your revised manuscript.

Kind regards,

Claudio M. Soares, Ph.D

Academic Editor

PLOS ONE

Reviewers' comments:

Reviewer's Responses to Questions

**Comments to the Author**

1. If the authors have adequately addressed your comments raised in a previous round of review and you feel that this manuscript is now acceptable for publication, you may indicate that here to bypass the “Comments to the Author” section, enter your conflict of interest statement in the “Confidential to Editor” section, and submit your "Accept" recommendation.

Reviewer #1: (No Response)

Reviewer #2: All comments have been addressed

2. Is the manuscript technically sound, and do the data support the conclusions?

Reviewer #1: Yes

Reviewer #2: (No Response)

3. Has the statistical analysis been performed appropriately and rigorously? 

Reviewer #1: Yes

Reviewer #2: (No Response)

4. Have the authors made all data underlying the findings in their manuscript fully available?

Reviewer #1: Yes

Reviewer #2: (No Response)

5. Is the manuscript presented in an intelligible fashion and written in standard English?

Reviewer #1: Yes

Reviewer #2: (No Response)

6. Review Comments to the Author

Reviewer #1: I am unable to accept this work for publication because the study closely replicates the work of Thomas et al (Cell Cycle 2010,9,4560-4568). It is already well-known that staple placement is important. It is also known that the two additional staple positions the authors tried are not optimal, based on experiments carried out by Verdine et al (JACS 2007) and simulations carried out by Thomas et al. The authors also fail to discuss the differences or advantages, if any, in their approach and results with the work of Thomas et al.

Reviewer #2: All comments have been addressed. I just note that the resolution of the figures should be improved before publication, and the phrase "major favorite contributor" in the abstract should be reworded for readability.

7. PLOS authors have the option to publish the peer review history of their article (what does this mean?). If published, this will include your full peer review and any attached files.

Reviewer #1: No

Reviewer #2: No

---

## [Author Response · Author response to Decision Letter 1]

12 Apr 2020

Our response to the reviewers' comments is described in the file "response_letter.pdf" uploaded.

---

## [Decision Letter · Decision Letter 2]

20 Apr 2020

Effect of linker on the binding free energy of stapled p53/HDM2 complex

PONE-D-19-35957R2

Dear Dr. Ham,

We are pleased to inform you that your manuscript has been judged scientifically suitable for publication and will be formally accepted for publication once it complies with all outstanding technical requirements.

With kind regards,

Claudio M. Soares, Ph.D

Academic Editor

PLOS ONE

Additional Editor Comments (optional):

Reviewers' comments:

Reviewer's Responses to Questions

**Comments to the Author**

1. If the authors have adequately addressed your comments raised in a previous round of review and you feel that this manuscript is now acceptable for publication, you may indicate that here to bypass the “Comments to the Author” section, enter your conflict of interest statement in the “Confidential to Editor” section, and submit your "Accept" recommendation.

Reviewer #1: All comments have been addressed

2. Is the manuscript technically sound, and do the data support the conclusions?

Reviewer #1: Yes

3. Has the statistical analysis been performed appropriately and rigorously? 

Reviewer #1: Yes

4. Have the authors made all data underlying the findings in their manuscript fully available?

Reviewer #1: Yes

5. Is the manuscript presented in an intelligible fashion and written in standard English?

Reviewer #1: Yes

6. Review Comments to the Author

Reviewer #1: (No Response)

7. PLOS authors have the option to publish the peer review history of their article (what does this mean?). If published, this will include your full peer review and any attached files.

Reviewer #1: No

---

## [Editor Report · Acceptance letter]

22 Apr 2020

PONE-D-19-35957R2 

Effect of linker on the binding free energy of stapled p53/HDM2 complex 

Dear Dr. Ham:

I am pleased to inform you that your manuscript has been deemed suitable for publication in PLOS ONE. Congratulations! Your manuscript is now with our production department. 

With kind regards,

on behalf of

Dr. Claudio M. Soares 

Academic Editor

PLOS ONE